# A Morphometric Approach to Understand Prokaryoplankton: A Study in the Sicily Channel (Central Mediterranean Sea)

**DOI:** 10.3390/microorganisms11041019

**Published:** 2023-04-13

**Authors:** Giovanna Maimone, Maurizio Azzaro, Francesco Placenti, Rodolfo Paranhos, Anderson Sousa Cabral, Franco Decembrini, Renata Zaccone, Alessandro Cosenza, Alessandro Ciro Rappazzo, Bernardo Patti, Gualtiero Basilone, Angela Cuttitta, Rosalia Ferreri, Salvatore Aronica, Rosabruna La Ferla

**Affiliations:** 1Institute of Polar Sciences, CNR ISP, Spianata S. Raineri 82, 98122 Messina, Italy; giovanna.maimone@cnr.it (G.M.);; 2Institute of Anthropic Impacts and Sustainability in the Marine Environment, CNR IAS, Via del Mare 3, 91021 Capo Granitola, Italy; 3Laboratory of Hydrobiology, Institute of Biology, Federal University of Rio de Janeiro (UFRJ), Av. Prof. Rodolpho Rocco 211, Rio de Janeiro 21941 617, Brazil; 4Institute of Microbiology Paulo de Góes, Federal University of Rio de Janeiro (UFRJ), Av. Carlos Chagas Filho 373, Rio de Janeiro 21941 902, Brazil; 5Department of Environmental Sciences, Informatics and Statistics, Ca’ Foscari Unversity of Venice Campus Scientifico, Via Torino 155, 30170 Venice, Italy; 6Institute of Anthropic Impacts and Sustainability in the Marine Environment, CNR IAS, Lungomare Cristoforo Colombo 4521, 90149 Palermo, Italy; 7Institute for Studies on the Mediterranean, CNR ISMED, Via Filippo Parlatore 65, 90145 Palermo, Italy

**Keywords:** prokaryotic cell size, cellular morphotypes, environmental factors, Sicily Channel

## Abstract

A new understanding of plankton ecology has been obtained by studying the phenotypic traits of free-living prokaryotes in the Sicily Channel (Central Mediterranean Sea), an area characterised by oligotrophic conditions. During three cruises carried out in July 2012, January 2013 and July 2013, the volume and morphology of prokaryotic cells were assessed microscopically using image analysis in relation to environmental conditions. The study found significant differences in cell morphologies among cruises. The largest cell volumes were observed in the July 2012 cruise (0.170 ± 0.156 µm^3^), and the smallest in the January 2013 cruise (0.060 ± 0.052 µm^3^). Cell volume was negatively limited by nutrients and positively by salinity. Seven cellular morphotypes were observed among which cocci, rods and coccobacilli were the most abundant. Cocci, although they prevailed numerically, always showed the smallest volumes. Elongated shapes were positively related to temperature. Relationships between cell morphologies and environmental drivers indicated a bottom-up control of the prokaryotic community. The morphology/morphometry-based approach is a useful tool for studying the prokaryotic community in microbial ecology and should be widely applied to marine microbial populations in nature.

## 1. Introduction

In aquatic systems, the shape and size of microbial cells are the main traits that affect their nutrient acquisition, light capture, grazing resistance, and vertical migration [1]. The shape and size of individual organisms are key parameters to characterize their ecology. Such trait-based studies suggest a significant ecological role in structuring the planktonic food web in relation to bottom-up/top-down controls [2]. In fact, in aquatic systems, protozoan predation is selective on the size and shape of prokaryotic cells (top-down pressure) [3], and bottom-up control affects the variability in microbial community phenotypic traits [4]. In addition, the dimensional characteristics of prokaryotic cells are of great importance for their use in estimating biomass using a conversion factor to transform cell numbers to carbon content. Hessen et al. [5] directly related temperature to community size structure, cell and genome size, formulating the empirical temperature–size rule. The forerunner of the cell-size approach, [6] explored the idea of using morphology as a proxy for function in phytoplankton ecology (functional groups). Acevedo-Trejos et al. [7] linked phytoplankton cell size to nutrient uptake, zooplankton grazing, and phytoplankton sinking. Sourisseau et al. [8] recently adopted trait-based studies for understanding the phenology of some toxic dinoflagellate species, the significance of environmental variability for species niches and competition for resources [9]. At the level of the prokaryotic community, the diversity of cell sizes can be associated with different populations [10] but, beyond taxonomy, the trait-based cell description concerns adaptive strategies to environmental factors [11,12]. The flexibility of prokaryotic cell traits provides an advantage in natural selection and ecological interactions [13,14]. Most studies of cell morphology and morphometry have been performed in the laboratory and mainly on pure cultures ([15] and references therein). A mechanistic model that takes into account the microbial ability to switch between different phenotypes was experimentally calibrated and confirmed on an isogenic population in chemostats [16]. Conversely, relatively few studies have been carried out on field populations. In this context, the cell shape undoubtedly co-varies with genome size, growth rate, protein synthesis rate and metabolic activity, as well as with functional diversity and trophic status [7,17]. 

In the frame of the European Marine Strategy Framework Directive (MSFD) of the EU’s marine waters, the determination of the prokaryotic biomass was proposed as a descriptor of biological diversity [17]. Estimation of trait-derived biomass can also be of predictive value for describing environmental variability in the scenario of current climate change [18]. Contradicting results have been obtained regarding the temperature-to-size relationship [5,19] as a prognostic tool associated with temperature changes, since causal relationships are not yet clear. 

The phenotypic traits of free-living prokaryotic cells have already been studied in several areas of the Mediterranean Sea [18,20,21,22], but there are still no quantitative data for the Sicily Channel (Central Mediterranean Sea), an important fishing area of the Mediterranean Sea. The present study aims to assess the suitability of the morphometric approach in this pelagic area thriving under oligotrophic conditions [23,24,25]. Here, it is assumed that hydrological conditions shape the macro- and microbiological parameters. An environment with low productivity imposes a challenge for microorganisms in sustaining their metabolic activity and population size; under these conditions, different phenotypes of relatively active “growing” and “non-growing” cells can coexist [16]. 

The specific objectives of this study were as follows: i) assess the prokaryotic cell-traits variability during three surveys conducted in the Sicily Channel, and ii) evaluate whether morphometrical and morphological traits were constrained by environmental pressures. For this purpose, image analysis on epifluorescence microscopy micrographs was applied to determine the size and shape of microbial cells. 

## 2. Materials and Methods

### 2.1. Study Sites and Sampling 

The surface circulation pattern in the Sicily Channel is characterized by a high dynamism and is locally controlled by the spreading of the Modified Atlantic Water (MAW), locally referred to as the Atlantic Ionian Stream (AIS; [26,27,28]). Table 1 shows the stations sampled during the cruises BANSIC-12 (BAN-12, 4–23 July 2012), NOVESAR-13 (NOV-13, 14–28 January 2013) and BANSIC-13 (BAN-13, 27 June–15 July 2013), all performed aboard the R/V “Urania”. 

Microbiological variables as well as hydrological and chemical parameters were collected throughout the water column (from surface to 100 m depth). Appendix A gives abbreviations for the studied parameters.

### 2.2. Environmental Variables 

Vertical profiles of temperature (T), salinity (S) and dissolved oxygen (DO) were obtained with a SeaBird 911plus CTD probe, and chlorophyll concentrations (FLUO) were estimated from fluorimeter AQUAtracka III (Chelsea Technologies Group) measurements. Specifically, chlorophyll concentration values were calculated by applying an empirical equation based on fluorescence calibration data and output voltage measurements by CTD. In accordance with the physicochemical discontinuities through the water column, identified by the CTD profiles, seawater samples were collected at three to four depths (from 5 to 100 m), including the deep chlorophyll maximum (DCM). 

With the exception of NOV-13, seawater samples (1.5–2.0 l) for the concentration of chlorophyll *a* (Chl *a*) were collected and filtered through Whatman GF/F glass-fiber filters (nominal pore size of 0.7 µm) and nuclepore polycarbonate membranes of 10.0 μm and 2.0 μm to obtain the following size fractions: micro- (≥10.0 μm), nano- (≥2.0 and <10.0 μm) and pico-phytoplankton (≥0.2 and <2.0 μm), respectively. Filters were stored in aluminium foil at −20 °C until laboratory analysis. Chl *a* was extracted with 90% acetone at 4 °C for 24 h in the dark and measured with a spectrofluorometer (mod. Varian Cary Eclipse) before and after acidification. The excitation and emission wavelengths (429 and 669 nm) were chosen after standardization with a Chl *a* solution, extracted from *Anacistys nidulans* (by Sigma Co., Milan, Italy). The Chl *a* concentration was calculated as described by Decembrini et al. [29].

All materials for dissolved nutrients analyses were preconditioned with 10% HCl and washed 2–3 times with ultrapure water. Unfiltered samples were stored in Falcon tubes (15 mL) at −20 °C. In the laboratory, the concentrations of nitrates (as N-NO_3_, hereinafter referred to as NO_3_), silicate (SiO_4_) and orthophosphate (P-PO_4_, hereinafter referred to as PO_4_) were measured using the Seal Autoanalyzer ‘‘QUAATRO’’ according to classical methods [30] adapted to the automated system. 

### 2.3. Microbial Variables

The studied prokaryotic variables were as follows: cell abundance (PA), volume (VOL), cell carbon content (CCC), biomass (PB) and cell morphotype. Detailed methodological procedures were reported in La Ferla et al. [31]. Briefly, seawater samples were immediately fixed with pre-filtered formaldehyde (0.2 µm porosity; final concentration 2%), and stored in the dark at 4 °C on board. According to Turley and Hughes [32], the frozen storage of freshly preserved and prepared seawater samples for up to 70 days results in no cell decrease. In our study, the slide preparation and cell counts were performed within 2 weeks from the sampling days in the laboratoty. Appropriate aliquots were filtered on polycarbonate black membranes (porosity 0.2 µm; GE Water & Process Technologies, Feasterville-Trevose, PA, USA) and stained with 4′,6-diamidino-2-phenylindole (DAPI, Sigma, final concentration 10 µg mL^−1^) according to Porter and Feig [33] for 20 min. Stained cells were counted using a Zeiss AXIOPLAN 2 Imaging microscope (magnification: Plan-Neofluar 100× objective and 10× ocular; HBO 100 W lamp; filter sets: G365 exciter filter, FT395 chromatic beam splitter, LP420 barrier filter) equipped with an AXIOCAM-HR digital camera. Micrographs were digitized on a personal computer using AXIOVISION 3.1 software (image acquisition at a standard resolution of 1300 × 1030 pixels). The pixel size of the resulting image was 0.106 µm with an automatic calibration. The system was further calibrated using fluorescent latex beads by measuring a FITC-stained suspension of monosized latex beads (diameter 2.13 µm). Cell recognition was performed by only one experienced operator, who discharged the misclassified objects from counting and measurement. The prokaryotic cell abundance (PA) was quantified on a minimum of 20 randomly selected fields in two replicate slides to minimize the analytical error and to minimize the effects of heterogeneous distribution. When the number of cells was very low, 30–40 random microscope fields were counted. In total, about 8800 cells were measured, and the volume (VOL, expressed in µm^3^) of each cell was calculated from two linear dimensions (width, W, and length, L), obtained manually. Detailed methodological procedures for single-cell volume calculation and volume-to-biomass conversion factors are reported in La Ferla et al. [31]. Image acquisition allowed us to determine the classification of morphotypes. Cells were operationally defined as cocci if their length (L) and width (W) differed by less than 0.10 μm, coccobacilli if their L and W differed by more than 0.10 μm, and rods if their L differed by at least two times, C-shaped and S-shaped cells were classified as vibrios and spirillae, respectively; cells larger than 4µm in L were defined as filamentous bacteria.

A few methodological considerations need to be made. Automated or semiautomated procedures are applied routinely for cell volume calculation by image analysis, adopting different protocols which take into account the cell-specific parameters [34]. Different algorithms are applied for the various dyes [35] and recently for different cell morphologies [36]. In the case of our study, the choice to follow the simplest, yet most used, procedure which takes into account only two dimensional parameters and the specific algorithm for DAPI dyeing was due to our need to compare our results with several previous data. A comment on the limits of DAPI staining for cell counting and volume estimates by Posch et al. [34] asserts that “DAPI has to be accepted as a usable stain at present”. 

With the exception of NOV-13, the flow cytometer has been adopted to evaluate virioplankton and prokaryotic cells as additional parameters associated with prokaryotic features. Virus abundance (VA)- expressed in virus-like particles (VLPs) was determined according to Brussaard [37], using a FACSCalibur flow cytometer (BD Biosciences) equipped with a 488 nm argon laser. In addition, prokaryotes (PA^C^) and sub-populations with different contents of nucleic acids (HNA, High Nucleic Acid, and LNA, Low Nucleic Acid) were determined according to Andrade et al. [34]. 

### 2.4. Statistical Analyses

Beanplots VOL, PA, and PB of the three sampling periods were generated using the beanplot package [38] in R (v. 2.14.2). This type of plot is an extended version of the well-known boxplots; in this case, the empirical distribution of the data is was also shown. 

Non-parametric analysis of variance (Kruskal–Wallis test) was applied to some variables to evaluate the differences among periods.

The Shannon index of diversity (H’) was applied to data on the abundance of specific morphological forms in the studied periods, using the Primer 6 software [39]. All statistical analyses were performed using R software [40,41]. Principal component analysis (PCA) was performed using the FactoMineR package, while the ggplot2 and factoextra packages were used to display PCA results [42,43,44]. The aim of PCA was to investigate possible relationships between microbiological variables (PA, PB, VOL and morphotypes) and environmental variables in three cruises. Before the PCA analysis, the data were transformed (log10x + 1).

## 3. Results

### 3.1. Hydrological and Chemical Features 

The hydrological and chemical features of the studied areas during the three cruises are reported in Table 2 (as ranges, averaged values and standard deviations).

During BAN-12 and BAN-13, the S minimum was recorded between 20 and 30 and 30 and 40 m, respectively. The mixed layer depth (MLD) developed at the depth of 10–15 m during BAN-12 and BAN-13, and in NOV-13, it affected almost the entire water column. Low DO was recorded in surface water in BAN-12, while the highest value was recorded in NOV-13. FLUO, estimated from probe measurements, was low in the surface waters of BAN-12 and BAN-13, when the DCM developed at a depth of about 60 m. In contrast, high FLUO values were recorded through the entire water column in NOV-13 (Patti et al. cruise reports). In general, FLUOs in BAN-13 accounted for almost a half and a third of those in BAN-12 and NOV-13, respectively (Table 2). The total Chl *a* concentration, measured fluorimetrically in the laboratory, displayed very low values, close to the limit of detection. In BAN-13, the average content of total Chl *a* was almost a half of the average values in BAN-12 (Appendix A ). In BAN-12 and BAN-13, fractionated Chl *a* (by the pore size filtration in laboratory) showed that the pico-sized fraction accounted for 64% and 51% of the total phytoplankton biomass, the nano-sized fraction for 25% and 30%, and micro-sized fraction for the 10% and 18% of the total phytoplankton biomass, respectively (Decembrini Unpublished Work).

The vertical distribution of nutrient concentrations in the three surveys showed that surface water was poor in nutrients, while the highest concentrations were measured in the deep layers (Figure 1). 

The main differences were evidenced in NO_3_ and SiO_4_ in the measurements carried out in NOV-13 (winter) compared with BAN-12 and BAN-13 (summer). In particular, in NOV-13, surface waters (0–30 m depth) were characterized by approximately 1 µmol L^−1^ of NO_3_ and SiO_4_ (Figure 1d,f) while in BAN-12 and BAN-13 this concentration reached depths of more than 60 m (Figure 1a,c,g,i).

### 3.2. Microbial Variables

The range of variation, mean and standard deviation of PA, VOL, CCC, PB and VA are reported in Appendix A. PA was in the order of 10^6^ cells ml^−1^ with the lowest values in NOV-13 and the highest in BAN-12. VOL exhibited a strong variability among cruises (Kruskal–Wallis test: *p* < 0.001) with the smallest and largest cell sizes in NOV-13 and BAN-12, respectively. Accordingly, CCC varied widely among surveys, with a global mean value of 36.2 ± 31.8 fg C cell^−1^. Appendix A shows the corresponding cellular carbon contents (CCC) for each sample and depth used to quantify the biomass. PB, being modulated by both PA and CCC, was 6.4 and 3.6 times higher in BAN-12 and BAN-13 than in NOV-13, respectively. In Figure 2, the beanplots show the distribution of the mean values of VOL, PA and PB in each survey. In BAN-12 and BAN-13, most of VOL was distributed over two well-defined cores, showing that the bimodal distribution and the mean values exceeded and were close to the annual average, respectively. In NOV-13, cell sizes showed a unimodal distribution that fell completely below the year-to-year average. PA depicted a unimodal distribution and homogeneous patterns in NOV-13 and BAN-13, while in BAN-12 an uneven core depicted cell size distribution. Only in NOV-13, PB turned out to be to be inscribed in a homogeneous core.

PA^C^ was in the order of 10^5^ cell ml^−1^ with higher values in BAN-13. The low nucleic acid (LNA) populations always overwhelmed the high nucleic acid (HNA), (HNA/LNA ratios 0.6 and 0.9 in BAN-12 and BAN-13, respectively).

VA was in the order of 10^5^ VLP ml^−1^ (Appendix A). Three different viral sub-population groups (V1, V2 and V3) were distinguished based on their fluorescence signals. In BAN-12, V1 group (mainly bacteriophages) appeared to be quantitatively predominant, accounting for 89% of the total VA, the V2 group accounted for 11% of the total, while the V3 group was negligible. The V2 group was more abundant (45%) in BAN-13, followed by the V1 (38%) and V3 groups (17%). In any case, the VPRs ratios (virus to prokaryote ratio) were low (0.5 and 1.4 in BAN-12 and BAN-13, respectively).

### 3.3. Cell Morphotypes and Their Morphometry

The cell VOL, CCC and biomass of each morphotype are reported in Figure 3A and Figure 3B, respectively. 

Despite the low abundance, vibrios and spirillae exhibited the largest VOL, mostly in BAN-12. Conversely, cocci showed the smallest VOL, particularly in NOV-13 (0.039 ± 0.037 µm^3^); coccobacilli, rods and curved rods were intermediate. CCC and the biomass (as percentage of the total biomass) of each morphotype are shown in Figure 3B. In BAN-12 and BAN-13, vibrios and spirillae accounted for the highest CCC value (>60 fg C cell^−1^), while the other morphotypes ranged from 25 to 54 fg C cell^−1^. In NOV-13, with the exception of cocci, which showed a low CCC (14 fg C cell^−1^), the different morphotypes showed similar CCC (mean value 25 fg C cell^−1^). The greatest weight in terms of biomass was found in BAN-12 when rods accounted for the 52% of the total biomass, followed by cocci and curved rods (42 and 37%, respectively). Despite their discrete size, vibrios and spirillae accounted barely for the 4.5 and <2% of the total biomass.

The morphometric data on prokaryotic types are summarised in Table 3. 

The cell L and W showed differences among the cruises. In BAN-12, the highest morphometrical measurements occurred, and in NOV-13, the smallest ones. Cocci averaged 0.46 μm in diameter with smaller cells in NOV-13. Elongated forms showed smaller (1.31 × 0.32 μm) and larger (1.68 × 0.41 μm) cells in NOV-13 and BAN-12, respectively. Within the curved cells, wide ranges in L and W were observed, with the smallest ones measured in NOV-13. Filamentous forms were found in BAN-13 only and were relatively short, showing an average L of 3.37 ± 1.54 µm. 

The morphological index (M = % rods/%cocci) always resulted in <1 and reached the maximum in NOV-13 (M = 0.9) and the minimum in BAN-13 (M = 0.44). 

Different classes of the microbial cell sizes were detected in the cruises: in BAN-12, the >0.20 µm^3^ size-class contributed 32 % of the total while, in NOV-13, the 0.02–0.049 µm^3^ made up 35%, and in BAN-13, the size-class 0.02–0.049 µm^3^ and 0.05–0.079 µm^3^, each represented 22% of the total. Cell length classes were mainly represented by cells belonging to the 0.4–0.8 µm class, accounting for 46% of the total number of cells, followed by the 0.8–1.2 µm class, accounting for 18% of the total cell number. 

### 3.4. Relations between Prokaryotic Features and Environmental Variables 

The relationships between prokaryotic features (PA, PB and VOL) and environmental variables across the three cruises are reported in Figure 4. The biplot displays the variables and their relationships in the space represented by the first two principal components. Variables with arrows placed closer together had a higher correlation. Positively correlated variables pointed to the same side of the plot. Negatively correlated variables indicated opposite sides of the graph. Individuals with a similar profile are grouped together.

According to the contribution of the variables on the two main components of the PCA, the 12 variables were grouped into four groups formed by T, NO_3,_ SiO_4_, FLUO and depth; S and DO; PB, PA and VOL; and PO_4_ only_._

The first two axes of a PCA analysis based on the abiotic data accounted for 78.3% of the total variance (PC1 = 46.8%, PC2 = 26.1%).

From the examination of the points relating to the various stations and position of the average value of each cruise, a defined ordering of the cruises through the first main component (displayed on the PC1-axis) was evidenced. The NOV-13 samples were higher than other samples on this component, mainly regarding the NO_3,_ FLUO and SiO_4_ variables. 

The samples of BAN-12 and BAN-13, characterized by lower values of NO_3_ and SiO_4_ and higher T values than NOV-13, generally fell in the middle of the plot, although some overlap in the ordering of the stations of them through the first main component were noted.

PA, PB and VOL were inversely correlated with PO_4_ and slightly less with NO_3_ and SiO_4_, while low positive correlations with S and depth were observed.

The relationships between the abundance of each morphotype and environmental variables in the three cruises are shown in Figure 5. According to their contribution in the two main components of the PCA, the variables were grouped into four groups formed by T, spirillae, vibrios, curved rods, NO_3_, SiO_4_ and depth; FLUO, PO_4_ and cocci; S and DO; rods and coccobacilli.

The first two axes of a PCA analysis based on the abiotic data accounted for 76.3% of the total variance (PC1 = 44.8%, PC2 = 31.5%). In this case BAN-12 and BAN-13 also showed a similar variability, more evident in comparison with NOV-13, characterized by the low presence of spirillae, vibrios and curved rods that strongly correlated positively with T and negatively with FLUO, and depth.

Finally, the high variability in the number of morphotypes compared to different cruises was also shown by the Shannon diversity index values (H’), being high in BAN-12 (H’ range: 2.57–3.22), low in NOV-13 (range: 2.1–2.3) and intermediate in BAN-13 (range: 2.3–2.7). 

## 4. Discussion

Understanding and explaining microbial community structures in response to environmental parameters and gradients are is a central goals in ecology. The study of cell traits improves the predictability of community shifts in response to changing environmental conditions and opens up new perspectives for a better understanding of the ecological properties of the ecosystem [45]. Moreover, the relationship between the structure of the prokaryoplankton and environmental factors can provide a theoretical tool for environmental protection and restoration of aquatic systems [46]. 

In this study, environmental parameters evidenced highly stratified systems in BAN-12 and BAN-13 (ΔT~10 °C) and a well-mixed system in NOV-13 (ΔT < 2 °C). Clear differences in DO and FLUO distribution with depth were evidenced, with expected seasonal variations. According to Bonanno et al. [47], the S minimum off the Sicilian coast represented a signal of the AIS on its typical trajectories for the considered periods. In BAN-13, the AIS signal, when moving eastwards and approaching the Ionian Sea, was weak, probably due to the presence of the ISW or due to the presence of an anticyclonic circulation located in the eastern part of the Sicilian coast, supporting the spread of the AIS vein off the southern Sicilian coast [48]. 

As concerns salinity, it was positively correlated to VOL, in contrast to what was found by La Ferla et al. [17] in several ecosystems characterized by different trophic states, but in accordance with Gocke et al. [49] in a study carried out in a hypertrophied tropical lagoon of Colombia. Since S is an index of dilution by fresh-water input, it is reasonable to assume that, in our samples, VOL variability was an indirect response to the input of Atlantic waters. Temperature is considered to be a fundamental parameter influencing the activity and growth of aquatic bacteria. In our case, a relation with T was not evidenced, suggesting that other environmental factors ruled out the morphometric variability. It is likely that water temperature acted simultaneously with other factors (the quality and quantity of organic nutrients, phytoplankton activity, flagellate grazing), controlling microbial dynamics [50]. In any case, contrary to deterministic temperature–size rules, which predicted larger body size or genome size in cold environments than at high temperatures [16,51,52], it cannot be excluded that in winter, VOL might have been sensitive to thermal stress as seen in riverine and deep-sea environments [2,21]. 

Lower nutrient concentrations in the surface layers of the Sicily Channel than previous estimates, and in some cases even lower than the detection limits of the analytical procedure, asserted the trophic state of the study areas. According to Schroeder et al. [53], nutrients concentration increased with depth in BAN-12 and BAN-13, where the nutrient patterns were similar [24]; in NOV-13, the difference between the estimates of surface and bottom waters was small (about 1 µmol L^−1^ of NO_3_). The extremely low nutrients and Chl *a* concentrations in combination with the predominance of the pico-sized phytoplankton confirmed the oligotrophy of the study area [54], mainly in BAN-13. Later, very low ratios of virus to prokaryotic (VPR: 1.36 and 0.52 in BAN-13 and BAN-12, respectively) were determined in this study. This ratio is considered as an indicator of a P-limited prokaryotic activity, and thus showed that the prokaryotic activity was insufficient to maintain a high level of virus production as already reported in the Sicily Channel [54]. Altogether, the relationships between cell VOL and physical–chemical parameters depicted the following scenario: the decrease in salinity, rather than the increase in temperature, appears to have a direct effect on the increase in prokaryotic biomass. Taking into consideration the already established oligotrophy of the studied area, it could be hypothesized that, in a climate changing scenario, an imbalance towards heterotrophic microbial populations to the detriment of the productive fraction could cause damage to commercial exploitation in fishing-related activities. 

Cell volume calculation has proven to be a useful approach to better quantify biomass and describe cell heterogeneity in mixed assemblages [55]. Since 2002, studies on prokaryotic cell measurements have been carried out in several areas of the Mediterranean Sea, mainly in relation to biogeochemical processes [18,20,21,56]. Generally, cell VOL increased with depth, and in the Ionian Sea, the cell VOL was useful to differentiate the main water masses spreading across the Mediterranean Sea [22]. Prokaryotic cell shape and size were considered appropriate parameters for characterizing trophic ecosystems [17], since small cells (VOL < 0.1 µm^3^) characterized oligotrophic systems, while large cells (VOL > 0.3 µm^3^) characterized eutrophic ones. Quantitative data on the size and morphology of prokaryotes have so far been missing for the Sicily Channel, with the exception of the study by Zaccone et al. [57]. These authors showed that the prokaryotic biomass, defined by cell volume, was supported by enzymatic activities. In our study, cell size was in a wider range than in other areas of the Mediterranean Sea [20]. A shift in the community structure in terms of VOL was observed with larger cell sizes in both warm periods (BAN-12 and BAN-13) than in the cold periods (NOV-13). The HNA/LNA ratios also confirmed the predominance of large VOL cells mainly in BAN-12. 

Probably, in response to changing environmental conditions, prokaryoplankton developed different survival strategies. The community showed larger cell size during periods of nutrients deficiency (K-strategy). The nutrient limitation in summer had led to the development of more complex nutrient acquisition strategies in agreement with Wentzky et al. [45]. Regarding nutrients, according to Teixeira et al. [58], the observed negative correlation between NO_3_ vs. PA and PB may depend on the prokaryoplankton efficiency in assimilating nutrients at low concentrations. A study on *Caulobacter crescentus* [54] stated that phosphate-poor conditions induced the formation of large bacterial cells. The Sicily Channel was characterized by a limitation of phosphorus [23] and in present study, large size cells were found in summer cruises. In the same area, high alkaline phosphatase activity was observed contributing to an efficient incorporation of C into the prokaryotic biomass in winter [57]. Conversely, in NOV-13, prokaryotic growth was probably more limited by carbon availability, and a predominance of coccoid cells was observed according to Sjöstedt et al. [50] and Sigee [59]. 

Considering planktonic dynamics, small prokaryotic cells seem to consume only the labile fraction of organic substrates produced by algae, while the larger ones seem to use more refractory sources of nutrients [60]. In our study, no relation linked cell size to phytoplankton population, at least in terms of Chl *a*, suggesting no potential limitation to prokaryotic cell size by labile organic matter of new production. However, the size diversity of prokaryoplankton temporally changed together with the morphotypes’ composition, in agreement with the seasonal reorganization of phytoplankton communities in temperate regions [7]. Quiroga et al. [61] explained the “miniaturization” of the cells as due to the interplay of nutrient limitation and size-selective protistan predation. A well-known topic is the shift from ‘top-down’ to ‘bottom-up’ controls on the composition of the microbial assemblage [62]. According to Pernthaler and Amann [3], large prokaryotic cells were preferred for predation. In our study, in relation to this conceptual model, the cellular lengths suggested the exclusion of the flagellate’s ingestion and the top-down control appeared to be weak or lacking. Relatively few studies have so far dealt with prokaryotic shape in relation to the environmental parameters. In the present study, the morphotypes’ structure appeared to be partially affected by environmental conditions and, in particular, vibrios, spirillae and curved rods correlated with environmental factors. 

A remarkable variation of morphotypes’abundance among the cruises was observed (ANOVA *p* < 0.0001). As referred by Posch et al. [34], single or multiple nutrient limitations can strongly affect the morphological variability of single strains, e.g., of *Vibrio* sp. or *Caulobacter* sp. Regarding field studies, grazing pressure and nutrient limitation might influence the in situ composition of morphotypes and their activities [62]. In this context, the high surface-to-volume ratios of small cells could be due to an adaptive strategy to low nutrient concentrations in aquatic systems [13]. 

In BAN-12, the positive relationship between cocci abundance and the pico-sized Chl *a* concentration (r = 0.477, *p* < 0.01, *n* = 28) lets us assume that the picophytoplankton could be the most abundant population. Conversely, in BAN-13, rods were negatively correlated with pico-sized Chl *a* concentration (r = −0.493, *p* < 0.05, *n* = 16) as already observed by Racy et al. [63] in a system with low resource availability. 

## 5. Conclusions

The starting hypothesis that environmental characteristics modulated prokaryotic cell size and shape in the studied area of the Sicily Channel was supported by our results. The trait-based approach suggests that the local conditions assumed a relevant role in shaping the morphometrical and morphological structure of prokaryotic community at least on a time scale. In any case, the prokaryotic traits were affected by the general oligotrophic conditions of the area, suggesting a bottom-up control. In fact, the variability of size and shape of micoorganisms did not seem to interact with larger organisms. The changes in prokaryoplankton structure were directly and/or indirectly dictated by the environmental variables that acted in concert and provoked complex combinations of responses. Although our dataset is not sufficiently quantitatively robust to draw general conclusions, the phenotypic approach appears to be a promising tool to understand community dynamics of the study area in response to changing environmental conditions.

## Figures and Tables

**Figure 1 microorganisms-11-01019-f001:**
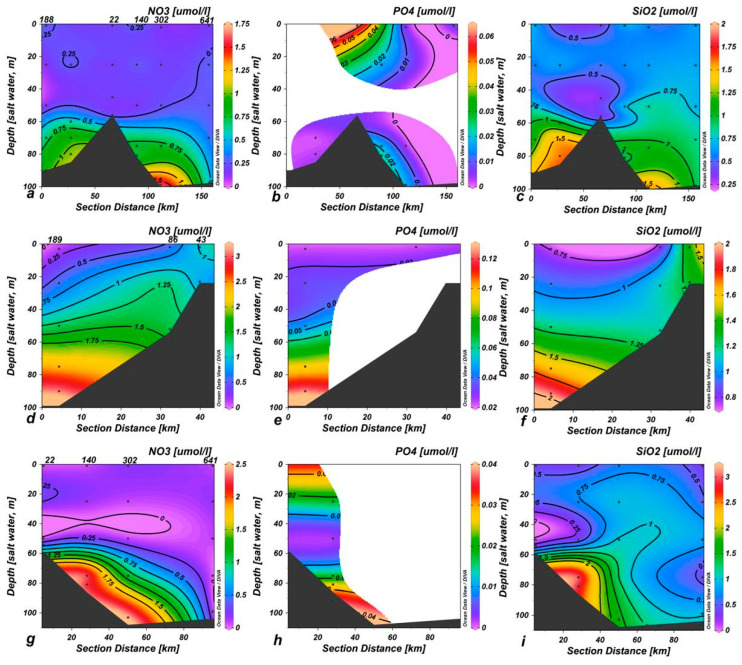
Vertical sections of nitrates, phosphates and silicates (µmol L^−1^) related to different oceanographic cruises. Specifically, section (**a**–**c**) refer to BANSIC 2012; (**d**–**f**) to BANSIC 2013; (**g**–**i**) to NOVESAR 2013.

**Figure 2 microorganisms-11-01019-f002:**
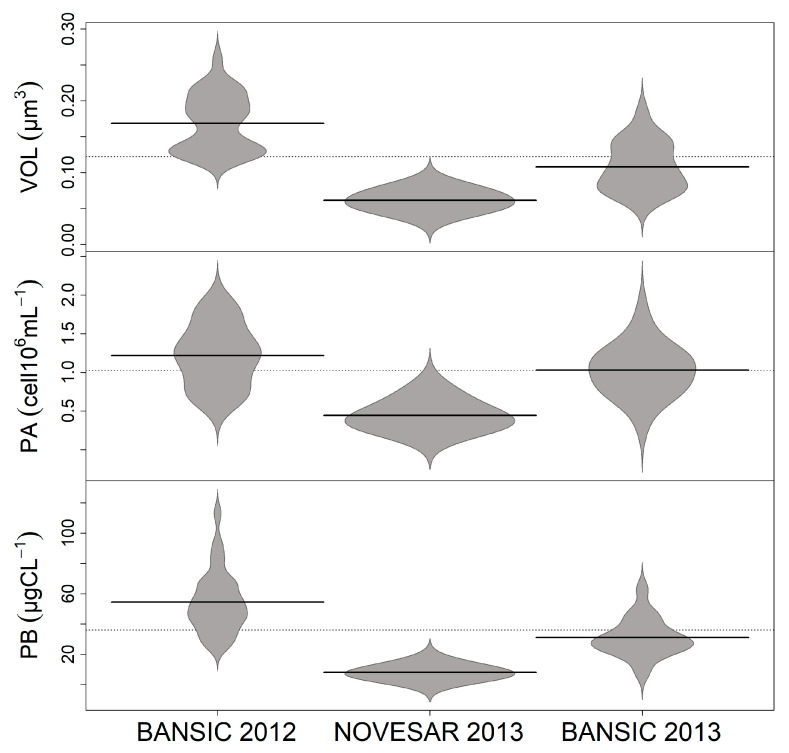
Beanplots of the entire dataset of the volumes (VOL), abundance (PA) and biomass (PB) of prokaryotic cells distributed in BANSIC 2012, NOVESAR 2013 and BANSIC 2013. Dashed lines represented overall mean values, black lines mean values within each period and grey areas show the empirical distribution of each parameter.

**Figure 3 microorganisms-11-01019-f003:**
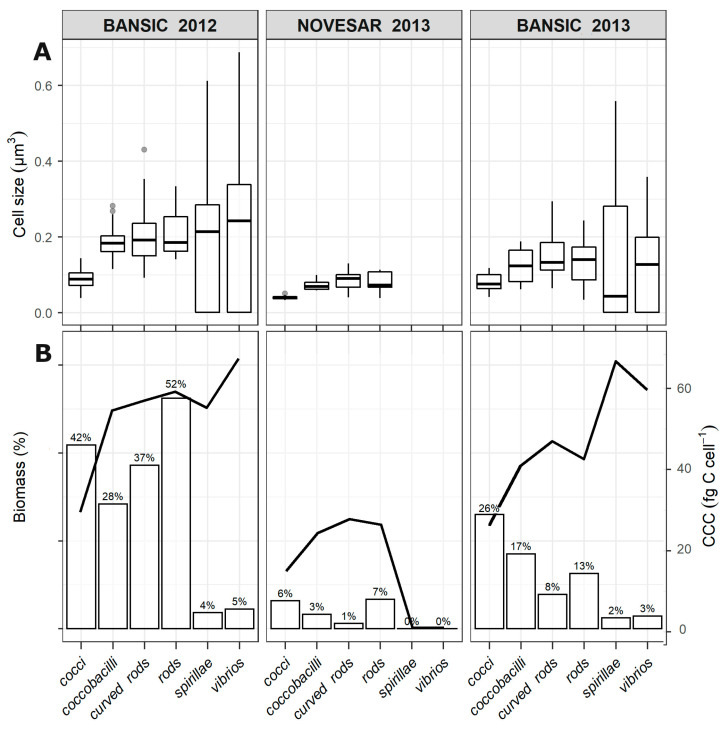
Cell volume of the morphotypes (µm^3^) (**A**) and CCC together with the relative biomass (as percentage of the total) of each morphotype (**B**) in BANSIC 2012, NOVESAR 2013 and BANSIC 2013.

**Figure 4 microorganisms-11-01019-f004:**
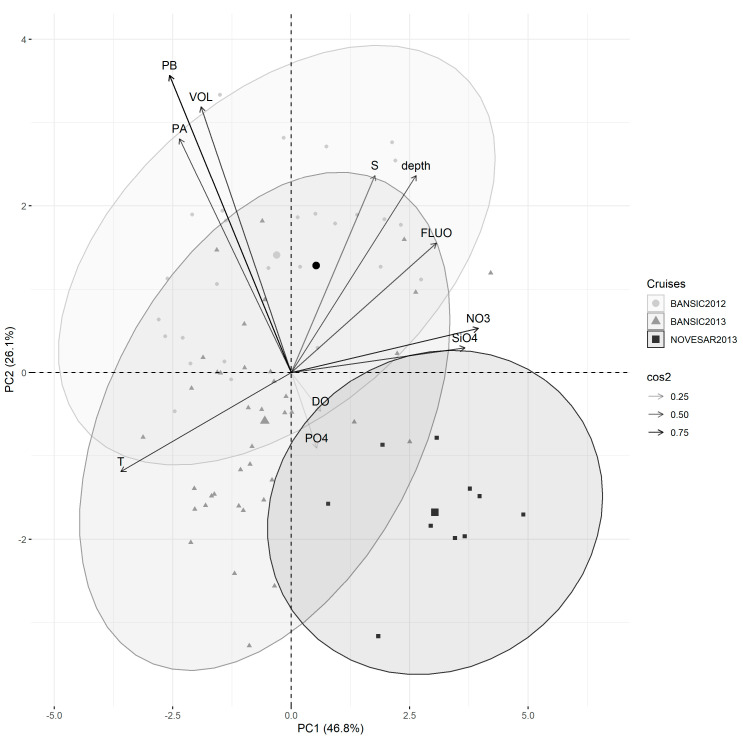
PCA ordination biplots for the microbiological and environmental variables analyzed in BANSIC 2012, NOVESAR 2013 and BANSIC 2013. The percentage of variance explained by each represents the squared loadings for variables and shows the importance of a principal component for a given observation. Filled circles represent samples collected from BANSIC 2012 cruise; filled triangles represent the samples collected from BANSIC 2013 cruise; filled squares represent the samples collected from NOVESAR 2013 cruise.

**Figure 5 microorganisms-11-01019-f005:**
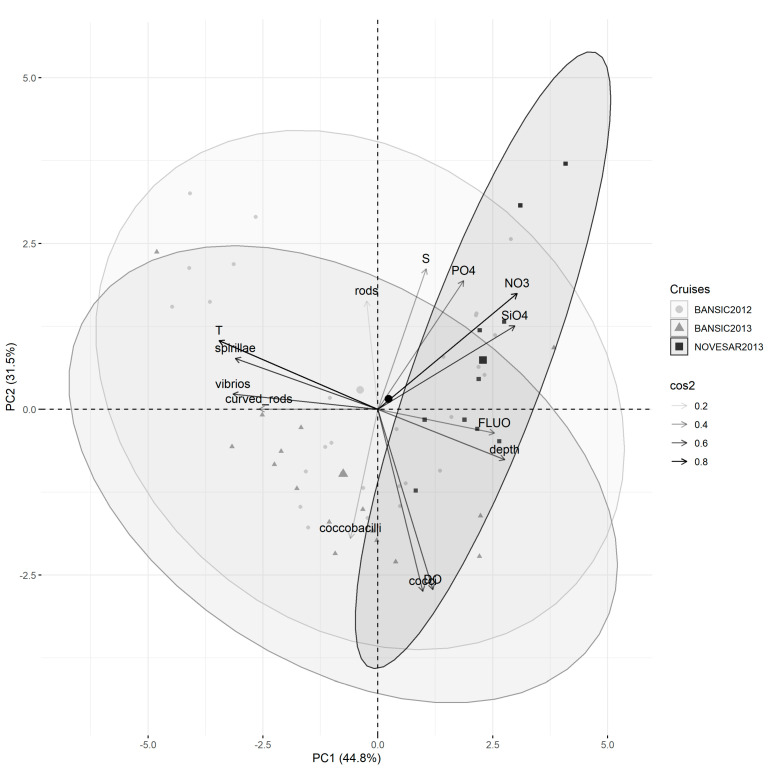
PCA ordination biplots for the morphotypes data and environmental variables analyzed in BANSIC 2012, NOVESAR 2013 and BANSIC 2013. The percentage of variance explained by each component is given between brackets. Arrows represent the variables included in the analyses. cos2 represents the squared loadings for variables and shows the importance of a principal component for a given observation. Filled circles represent samples collected from BANSIC 2012 cruises; filled triangles represent samples collected from BANSIC 2013 cruises; filled squares represent samples collected from NOVESAR 2013 cruises.

**Table 1 microorganisms-11-01019-t001:** Sampling stations, coordinates, names of cruise and measurements taken (x).

Cruise	Station	Longitude	Latitude	CTD	Microbiological	Dissolved
		[East]	[North]		Parameters	Nutrients
BANSIC-2012	22	15.235	36.638	x	x	x
	143	15.315	36.364	x	x	x
	137	14.886	36.565	x	x	x
	188	14.624	36.607	x	x	x
	302	14.967	36.292	x	x	x
	641	14.700	35.945	x	x	x
NOVESAR-2013	22	15.046	36.647	x	x	x
	137	15.001	36.589	x	x	x
	188	14.696	36.574	x	x	x
BANSIC-2013	2	15.259	36.787	x	x	
	22	15.232	36.638	x		x
	137	14.884	36.564	x	x	
	140	15.099	36.464			x
	296	14.535	36.491	x	x	
	641	14.699	35.945			x
	303	15.038	36.257	x	x	
	302	14.965	36.290			x
	461	14.188	36.418	x	x	x

**Table 2 microorganisms-11-01019-t002:** Ranges of variation, means and standard deviations of temperature (T °C), salinity (S), dissolved oxygen (DO), density (DEN), fluorescence (FLUO), nitrates (NO_3_), phosphates (PO_4_), silicates (SiO_4_) and total and fractionated Chl *a*. n.d. = not determined.

		BANSIC-12	NOVESAR-13	BANSIC-13
T °C	range	14.80–26.90	14.80–15.10	14.43–24.58
	mean ± sd	18.20 ± 4.40	14.90 ± 0.08	19.65 ± 3.75
S	range	37.3–38.70	38.0–38.80	37.56–38.71
	mean ± sd	38.3 ± 0.30	38.4 ± 0.25	38.10 ± 0.42
DO (mg L^−1^)	range	6.30–8.10	6.90–8.20	7.04–8.25
	mean ± sd	7.30 ± 0.60	7.70 ± 0.38	7.64 ± 0.46
DEN (kg/m^3^)	range	25.2–28.80	28.4–28.90	25.50–28.67
	mean ± sd	27.70 ± 1.29	28.6 ± 0.18	27.08 ± 0.99
FLUO (µg L^−1^)	range	0.02–0.39	0.04–0.38	0.01–8.25
	mean ± sd	0.13 ± 0.11	0.24 ± 0.12	0.07 ± 0.09
NO_3_ (µmol L^−1^)	range	0.13–1.62	0.15–3.12	0.03–2.43
	mean ± sd	0.45 ± 0.37	1.26 ± 0.90	0.38 ± 0.62
PO_4_ (µmol L^−1^)	range	0.001–0.070	0.023–0.127	0.02–0.02
	mean ± sd	0.018 ± 0.02	0.057 ± 0.042	0.02 ± 0.00
SiO_4_ (µmol L^−1^)	range	0.2–1.58	0.74–1.952	0.32–3.18
	mean ± sd	0.85 ± 0.36	1.22 ± 0.37	0.81 ± 0.69
Total Chl *a* (mg m^−3^)	range	0.024–0.317	n.d.	0.009–0.224
	mean ± sd	0.099 ± 0.077	n.d.	0.055 ± 0.056
micro-sized (>10 μm)	range	0.001–0.062	n.d.	0.001–0.056
	mean ± sd	0.012 ± 0.015	n.d.	0.011 ± 0.014
nano-sized (10–2.0 μm)	range	0.006–0.143	n.d.	0.003–0.071
	mean ± sd	0.026 ± 0.028	n.d.	0.016 ± 0.019
pico-sized (2–0.2 μm)	range	0.016–0.148	n.d.	0.004–0.097
	mean ± sd	0.065 ± 0.047	n.d.	0.029 ± 0.296

**Table 3 microorganisms-11-01019-t003:** Morphometrical dataset: average cell measures on total cells and per morphotype (*n* = the number of cells measured); averaged volumes and s.d. per each morphotype. L = Length; W = Width. n.d. = not determined.

		L (µm)			W (µm)		
		Mean ± Sd	Min	Max	Mean ± Sd	Min	Max
BANSIC-2012	Total cells	1.16 ± 0.79	0.32	6.34	0.47 ± 0.14	0.21	1.38
*n* = 4573	vibrios	2.52 ± 0.57	1.46	4.06	0.40 ± 0.11	0.21	0.64
	spirillae	2.90 ± 0.69	1.39	4.54	0.35 ± 0.09	0.21	0.64
	coccobacilli	0.85 ± 0.19	0.53	1.68	0.58 ± 0.12	0.32	1.05
	Cocci	0.51 ± 0.15	0.32	1.38	0.51 ± 0.15	0.32	1.38
	rods	1.68 ± 0.66	0.64	6.34	0.41 ± 0.10	0.21	0.76
	curved rods	1.99 ± 0.55	1.07	5.08	0.38 ± 0.12	0.21	0.81
	filamentous forms	n.d	n.d	n.d	n.d	n.d	n.d
NOVESAR-2013	Total cells	0.69 ± 0.42	0.32	3.88	0.37 ± 0.09	0.15	0.95
*n* = 1344	vibrios	n.d.	n.d.	n.d.	n.d.	n.d.	n.d.
	spirillae	n.d.	n.d.	n.d.	n.d.	n.d.	n.d.
	coccobacilli	0.62 ± 0.17	0.42	1.7	0.42 ± 0.08	0.32	0.95
	Cocci	0.40 ± 0.09	0.32	0.95	0.40 ± 0.09	0.32	0.95
	rods	1.01 ± 0.36	0.42	3.23	0.32 ± 0.08	0.15	0.65
	curved rods	1.38 ± 0.33	0.74	2.24	0.26 ± 0.07	0.15	0.44
	filamentous forms	n.d.	n.d.	n.d.	n.d.	n.d.	n.d.
BANSIC-2013	Total cells	0.85 ± 0.59	0.28	6.37	0.43 ± 0.14	0.11	1.83
*n* = 2852	vibrios	2.23 ± 0.75	1.14	6.37	0.36 ± 0.11	0.15	0.65
	spirillae	2.35 ± 0.73	0.71	3.94	0.34 ± 0.09	0.15	0.53
	coccobacilli	0.69 ± 0.18	0.28	2.19	0.46 ± 0.14	0.21	1.83
	cocci	0.48 ± 0.14	0.30	1.27	0.48 ± 0.14	0.30	1.27
	rods	1.17 ± 0.43	0.55	2.86	0.37 ± 0.11	0.11	0.65
	curved rods	1.56 ± 0.50	0.53	3.59	0.35 ± 0.10	0.53	3.59
	filamentous forms	3.37 ± 1.54	2.04	6.41	0.24 ± 0.012	0.21	0.24

## Data Availability

Not applicable.

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
