# Peer review of "A Morphometric Approach to Understand Prokaryoplankton: A Study in the Sicily Channel (Central Mediterranean Sea)"

_microorganisms, 2023, doi:10.3390/microorganisms11041019_

Round 1

Reviewer 1 Report

Overall, I thought the motivation for the study was well-justified. I have often emphasized to my students and colleagues that actual cell biomass (determined by size analysis and allometric relationships) is a more important variable than cell abundances alone, especially in the context of elemental fluxes in the environment. So congratulations for exploring this issue in detail.

I was a little disappointed, however, that you did not discuss the uncertainties involved in making biovolume estimates from epifluorescent images. For example, see Posch et al. 2001 (AME 25:55-63) discussion of the disagreement between DAPI and AO volume estimates and Turley's discussion of preservation and storage artifacts. I'm not saying that epifluorescent images are inadequate tools, but it is critical to assess the confidence that can be attributed to the analysis.  Some exploration of method's limitation would be most welcome.

Minor concerns are listed below.

First, I confess to being a little misled by the paper's title. I thought paper would explore phenotype or genotype traits rather than size and shape.  I suggest the following modified title:"A morphometric approach to understanding prokaryoplankton ecology: a study in the Sicily Channel (Central Mediterranean Sea)"

lines 79-81 - meaning is unclear.  Incomplete thought? 

line 80 - replace productivity-poor with "low productivity"

lines 145-146 - how about "two replicate slides to minimize the analytical error and to minimize the effects of heterogeneous distribution"?

Table 2 - somewhere please explain how FLUO sensor readings are converted to ug/L values as presented.

lines 188-189 - awkward phrasing.

line 219 - please specify what extrapolation factor was used.

lines 224-225. By "cores" do you mean "bins"?  This sentence is awkwardly phrased.

Lines 227-229 - difficult to understand meaning.4

line 236 - specify how two variables are ratioed.

line 243 - define "VPR" 

lines 249-250 - biomass contributions of each morphotype may as interesting or more so than numerical contributions.

Table 3. I only see L and W in table, no volumes as advertised.

lines 274-277 - difficult to understand your point.

line 317 - DENS and DEPTH necessarily covary. I don't see the point of including DENS in analyses and presentation.

lines 433-434 - seems like a bit of vague speculation.

lines 442-443 - also seems like a bit of vague speculative arm-waving.

Author Response

Overall, I thought the motivation for the study was well-justified. I have often emphasized to my students and colleagues that actual cell biomass (determined by size analysis and allometric relationships) is a more important variable than cell abundances alone, especially in the context of elemental fluxes in the environment. So congratulations for exploring this issue in detail.

-Thank you very much.

I was a little disappointed, however, that you did not discuss the uncertainties involved in making biovolume estimates from epifluorescent images. For example, see Posch et al. 2001 (AME 25:55-63) discussion of the disagreement between DAPI and AO volume estimates and Turley's discussion of preservation and storage artifacts. I'm not saying that epifluorescent images are inadequate tools, but it is critical to assess the confidence that can be attributed to the analysis.  Some exploration of method's limitation would be most welcome.

-Your suggestion is important. In the Materials and Methods, a section regarding the - methodological considerations has been included.

-According to Taylor (1992), the frozen storage of freshly preserved and prepared seawater samples for up to 70 days results in no cell decrease. In our study, the slide preparation and cell counts were done within 2 weeks from the sampling days.

Minor concerns are listed below.

First, I confess to being a little misled by the paper's title. I thought paper would explore phenotype or genotype traits rather than size and shape.  I suggest the following modified title:"A morphometric approach to understanding prokaryoplankton ecology: a study in the Sicily Channel (Central Mediterranean Sea)"

-Done

lines 79-81 - meaning is unclear.  Incomplete thought? 

-Sentence deleted

line 80 - replace productivity-poor with "low productivity"

-Done

lines 145-146 - how about "two replicate slides to minimize the analytical error and to minimize the effects of heterogeneous distribution"?

-Thank you for your suggestion

Table 2 - somewhere please explain how FLUO sensor readings are converted to ug/L values as presented.

-The explanation has been included in the section Materials and Methods

lines 188-189 - awkward phrasing.

-Sentence rearranged

line 219 - please specify what extrapolation factor was used.

-A supplementary table (Table S3) has been added where the CCC used are reported

lines 224-225. By "cores" do you mean "bins"?  This sentence is awkwardly phrased.

-The sentence has been rearranged

Lines 227-229 - difficult to understand meaning.

-The sentence has been rearranged

line 236 - specify how two variables are ratioed.

-Done

line 243 - define "VPR" 

-VPR = Virus-like particle abundance to Prokaryotic abundance by Flow Cytometer Ratio has been added in the text

lines 249-250 - biomass contributions of each morphotype may as interesting or more so than numerical contributions.

-Thanks for your suggestion. Figure 3B has been modified, showing the cellular carbon contents and related biomass (as percentage of the total biomass) of each morphotype

Table 3. I only see L and W in table, no volumes as advertised.

-The cell volumes have not been included in the table as they are shown in Figure 3

lines 274-277 - difficult to understand your point.

-The sentence has been rearranged

line 317 - DENS and DEPTH necessarily covary. I don't see the point of including DENS in analyses and presentation.

-OK, according to your right suggestion, DENS has been deleted

lines 433-434 - seems like a bit of vague speculation.

-You are right! This speculative sentence has been deleted

lines 442-443 - also seems like a bit of vague speculative arm-waving.

-You are right! This vague sentence has been deleted

Reviewer 2 Report

This research paper is based on well defined hypothesis considering that environmental characteristics modulated influences on cell size and body shape. The results and discussions are supporting aspects of this relationship. The authors are highlighting the fact that trait-based examination suggested that the local conditions are playing significant role in shaping the morphological structure of prokaryotic community at least on a time scale.

The paper deserves publication and advances the interrelationship of biotic and non-biotic components in aquatic environment.

The findings are of interest, and I would suggest that since at the introduction section the climate change issues are well hypothesized, it would be of inertest for the scientific community (and not only) that under the conclusion section authors should address the potential way of influencing prokaryotic cell size and body shape.  Affecting cell size might be reflected within higher level of food webs in marine environment.

Author Response

This research paper is based on well defined hypothesis considering that environmental characteristics modulated influences on cell size and body shape. The results and discussions are supporting aspects of this relationship. The authors are highlighting the fact that trait-based examination suggested that the local conditions are playing significant role in shaping the morphological structure of prokaryotic community at least on a time scale.

The paper deserves publication and advances the interrelationship of biotic and non-biotic components in aquatic environment.

The findings are of interest, and I would suggest that since at the introduction section the climate change issues are well hypothesized, it would be of inertest for the scientific community (and not only) that under the conclusion section authors should address the potential way of influencing prokaryotic cell size and body shape.  Affecting cell size might be reflected within higher level of food webs in marine environment.

  • Our data are the first obtained on the morphometric approach of the prokaryoplankton in the Sicily Channel. The only reference to climate change scenario is paper n° 18 where several oceanographic cruises over a decadal basis were examined. This is not the case of our study where thre oceanographic cruises within two years performed, therefore the collected results, cannot have a predictive value in the context of climate change (i.e. decadal long term series should analysed to draw a robust conclusion). A sentence concerning as the potential effect of the cell size and shape variability within the marine food webs has been added.

Reviewer 3 Report

The volume and morphology of prokaryotic cells were microscopically  evaluated by Image Analysis and related to environmental conditions in this manuscript. This work is interesting and valuable. Minor revision is needed before it could be accepted.

1.The title can't reflect the research content of the manuscript, it should be focused on cell morphological characteristics or cell size. A cell-trait approach is very blurry and not very clear; prokaryoplankton ecology is too big.

2. "3.1 hydrological features" should be hydrological and chemical features

3. Discussion should be reworked and highlighted your novel finding.

Author Response

The volume and morphology of prokaryotic cells were microscopically  evaluated by Image Analysis and related to environmental conditions in this manuscript. This work is interesting and valuable. Minor revision is needed before it could be accepted.

1.The title can't reflect the research content of the manuscript, it should be focused on cell morphological characteristics or cell size. A cell-trait approach is very blurry and not very clear; prokaryoplankton ecology is too big.

- According to your advice and that of the Referee 1, the title has been modified in “A morphometric approach to understand prokaryoplankton: a study in the Sicily Channel (Central Mediterranean Sea)”

  1. "3.1 hydrological features" should be hydrological and chemical features

- Done

  1. Discussion should be reworked and highlighted your novel finding.

- The discussions have been reworked